# Concurrent Validity of Digital Measures of Psychological Dimensions Associated with Suicidality Using AuxiliApp

**DOI:** 10.3390/bs15070868

**Published:** 2025-06-26

**Authors:** Miguel Zacarías Pérez Sosa, Diego de-la-Vega-Sánchez, Sergio Sanz-Gómez, Adrián Alacreu-Crespo, Pedro Moreno-Gea, Pilar A. Saiz, Julio Seoane Rey, José Giner, Lucas Giner

**Affiliations:** 1Hospital Universitario Virgen Macarena, 41009 Sevilla, Spain; miguelzacarias88.ull@gmail.com; 2Departamento de Psiquiatría, Universidad de Sevilla, 41013 Sevilla, Spain; ssanz1@us.es (S.S.-G.); jginer@us.es (J.G.); lginer@us.es (L.G.); 3Departamento de Psicología y Sociología, Universidad de Zaragoza, 50009 Zaragoza, Spain; aalacreu@unizar.es; 4Psiquiatria, 30204 Cartagena, Spain; pmoreno@psiquiatria.com; 5Departamento de Psiquiatría, Universidad de Oviedo, 33003 Oviedo, Spain; frank@uniovi.es; 6Centro de Investigación Biomédica en Red en Salud Mental (CIBERSAM), Instituto de Salud Carlos III, 28029 Madrid, Spain; 7Instituto de Investigación Sanitaria del Principado de Asturias (ISPA), 33011 Oviedo, Spain; 8Servicio de Salud del Principado de Asturias (SESPA), 33011 Oviedo, Spain; 9Departamento de Psicología Social, Universidad de Valencia, 46010 Valencia, Spain; seoane@uv.es

**Keywords:** suicide, suicide prevention, risk assessment, digital technology, mHealth

## Abstract

Suicide is a major public health concern, and accurate risk assessment is essential for prevention. Slider-format questions offer a quick, intuitive, and accessible method to evaluate suicide-related dimensions. This study examines the reliability of slider-based items compared to standardized psychometric instruments when delivered via a mobile app. A total of 299 university students completed a digital self-report questionnaire using the AuxiliApp mobile platform. Participants answered validated scales assessing depression, psychological pain, suicidal ideation, anger, impulsivity, loneliness, and reasons for living, each presented in both traditional Likert and novel slider formats. Pearson correlations were used to evaluate the relationship between traditional and slider-based scores. All correlations were statistically significant (*p* < 0.001). Moderate correlations were found in most domains, including depression, psychological pain, suicidal ideation, loneliness, and key aspects of impulsivity and anger. Lower correlations appeared in subscales related to anger control and protective beliefs against suicide. Slider-based items demonstrated acceptable psychometric equivalence and concurrent validity compared to traditional scales. Their brevity and compatibility with mobile devices support their use in telehealth and digital mental health screening. While not a replacement for clinical evaluation, they may facilitate early detection and ongoing monitoring in at-risk populations.

## 1. Introduction

Suicide remains a critical global public health issue, accounting for a significant number of deaths worldwide. According to the World Health Organization (WHO), nearly 800,000 people die by suicide each year, representing approximately 1.4% of all deaths globally ([98]). This issue is particularly pressing among young adults ([101]). Numerous risk factors have been identified in association with suicidal behavior, including the presence of a mental disorder—87.3% of individuals who die by suicide are estimated to have been diagnosed with a mental illness prior to their death ([5])—as well as traits such as impulsivity, aggressiveness, and emotional instability ([14]; [19]; [49]).

Impulsivity—defined as a predisposition to act without forethought or planning, often resulting in adverse consequences for oneself or others—has been consistently associated with suicidal behaviors ([41]; [84]; [90]; [59]), including self-harm ([37]; [36]), suicide attempts ([6]; [17]; [18]; [52]), and deaths by suicide ([1]; [20]; [13]), particularly among younger populations ([1]; [6]). This association is robust across various populations, including psychiatric inpatients, those with substance use disorders, and individuals with trauma histories, and its effect is especially pronounced when measured close in time to suicidal behavior ([44]; [47]). Neurobiological links also highlight impulsivity and aggression as associated with altered brain function in regions responsible for emotion regulation, potentially serving as biological markers for suicide risk ([48]; [90]).

Anger has also been linked to suicidal behavior ([39]; [53]; [74]; [59]), especially in individuals with post-traumatic stress disorder ([22]; [99]; [47]) and among adolescents and young adults ([62]; [69]), particularly young males ([21]; [25]). Associations have also been observed in populations with borderline personality disorder ([26]; [57]; [87]; [31]) and major depressive disorder ([67]). While anger alone may be less specific to suicide risk than impulsivity, its presence is often elevated in those with mental health issues ([44]; [13]). Crucially, high levels of both anger and impulsivity can act synergistically, further increasing suicide risk beyond the effect of either trait alone, mediating the impact of stressful life events on suicide risk ([44]; [59]; [13]).

Other psychological dimensions related to suicide risk include psychological pain ([43]; [92], [93]), loneliness ([24]; [46]), and the absence of reasons for living ([15]; [76]). To identify individuals at elevated risk of suicide, clinicians and researchers rely on assessment tools that evaluate these dimensions. The Beck inventories, for example, have been used to detect individuals at risk through scales assessing depression ([32]; [94]), hopelessness ([10]; [77]; [82]; [88]), and suicidal intent ([38]; [86]; [89]; [103]).

Several strategies have demonstrated efficacy in the prevention of suicidal behavior. These include early detection of suicidal ideation and behavior, the implementation of individualized safety plans during crises, timely access to specialized mental healthcare—especially following critical events such as hospital discharge—continuity of care, and the presence of strong social and family support networks ([95]; [102]). The COVID-19 pandemic appears to have exacerbated suicide risk through two primary mechanisms: by intensifying known risk factors such as psychological distress, social isolation, and hopelessness; and by undermining the functioning of public health systems ([23]; [73]). In response to these systemic strains and growing needs within the population, it is imperative to reconsider and innovate our approaches to suicide prevention.

New technologies and telemedicine are rapidly transforming healthcare delivery, offering unprecedented opportunities to reach individuals who might otherwise lack access to mental health services ([42]; [50]; [63]). The COVID-19 pandemic significantly accelerated the adoption of telehealth, demonstrating its feasibility and utility for a wide range of psychiatric interventions, including psychotherapy, medication management, and crisis support ([40]; [2]). Mobile applications, in particular, provide a ubiquitous and convenient platform for self-monitoring, psychoeducation, and even early detection of mental health concerns, including those related to suicide risk ([4]). These tools can bridge geographical barriers, reduce stigma, and offer discreet support, making them invaluable assets in modern mental health frameworks ([56]). However, it is important to note that a systematic review of mobile applications for depression and suicide prevention revealed that only a small fraction incorporated evidence-based strategies to address suicidal behavior ([91]). This highlights a critical gap in the application of digital technologies for suicide prevention.

Accurate and timely identification of individuals at risk is paramount for effective prevention strategies. Traditional methods of psychological assessment, though well-established, can sometimes be lengthy, resource-intensive, or less engaging, particularly for younger, digitally native populations. In this context, the development and validation of innovative assessment tools adapted to digital platforms are crucial for enhancing accessibility, efficiency, and user engagement in mental healthcare. Numerous studies have shown that participants tend to feel more comfortable completing electronic questionnaires than traditional paper-based ones. Electronic formats enhance accessibility and optimize time efficiency, while demonstrating acceptable psychometric equivalence, provided that standardized protocols are followed and the tools are properly validated ([8]).

Visual analog scales (VAS), or slider-type questions, offer a continuous graphical format that allows individuals to indicate the position that best reflects their subjective experience. These scales are generally quicker, simpler, and more intuitive to complete. Research has demonstrated robust correlations between VAS and traditional measures in domains such as affective and psychotic symptoms. For example, Palmier-Claus and colleagues ([68]) validated an analog-format self-report tool named ClinTouch, comparing it with the PANSS and Calgary scales, finding moderate to strong associations for several parameters. Similarly, a review examining digital self-report questionnaires in patients with depression and mania ([27]) found that instruments evaluated in depression studies demonstrated good validity and reliability when compared with established clinical scales.

Given the nascent development and empirical investigation of slider-format questions for these specific psychological constructs in mobile health applications, this study aims to compare the psychometric equivalence and utility of traditional paper-based scales with matched slider-format questions assessing dimensions related to suicide risk, including suicidal intent, impulsivity, aggressiveness, anger, depression, reasons for living, loneliness, and psychological pain. This study adopts an exploratory research design to examine the psychometric equivalence between these novel digital items and established standardized scales, all administered within a mobile application.

## 2. Materials and Methods

This study employed a quantitative, cross-sectional, and correlational design aimed at assessing the psychometric equivalence of slider-format questions against established standardized scales when administered via a mobile application. Data collection was synchronous, meaning all data for each participant were collected at a single point in time.

### 2.1. Participants

A total of 299 university students from various universities across Spain participated in the study. The mean age of the sample was M = 19.86, SD = 1.86 years, and 71.2% of the participants were women. A total of 73.9% of participants were enrolled at the University of Seville, while the remaining 26.1% attended the University of Oviedo. Most participants (121) were studying medicine (50.5%), followed by nursing and physiotherapy (42.8%), and criminology (6.7%). The recruitment of participants was conducted using a non-probabilistic convenience sampling method, involving invitations distributed through university channels and social media across the country.

Inclusion criteria for participation were being a university student, being 18 years of age or older, being healthy volunteers, providing informed consent, having access to a mobile device with internet connection, and ability to comprehend and respond to questionnaires in Spanish. Exclusion criteria for participation were failure to complete all study questionnaires, providing inconsistent or invalid responses (e.g., straight-lining, obvious random responses).

The study was conducted entirely in Spanish.

### 2.2. Sample Size and Power Analysis

A formal power analysis was not conducted prior to data collection. This study adopted an exploratory research design to investigate the psychometric equivalence of a novel assessment format for suicide-related dimensions. The sample size was primarily determined by convenience and participant availability within the university setting, aiming to recruit a sufficient number of participants to explore the relationships between the measures. While a power analysis was not performed, the collected sample size of 299 participants was considered appropriate for detecting correlations of moderate effect size (r > 0.3) given the nature of the study and in line with typical sample sizes in instrument validation research.

### 2.3. Ethical Considerations

The study protocol, titled ‘Validación de escalas de valoración del riesgo de suicidio en versión analógica’, was reviewed and received a favorable dictamen from the Comité Coordinador de Ética de la Investigación Biomédica de Andalucía on 29 May 2017, under protocol version V1. All participants provided informed consent digitally prior to their participation. They were fully informed about the study’s objectives, procedures, potential risks, and benefits, as well as their right to withdraw at any time without penalty. Participant data were handled with strict confidentiality and anonymity, in accordance with ethical guidelines.

### 2.4. Procedure

The study was conducted entirely through AuxiliApp, a proprietary mobile application designed for mental health assessment and support. All participants completed a self-administered digital questionnaire via this mobile platform. Data collection was therefore 100% digital, ensuring consistency in the administration format.

Regarding the administration flow, participants first completed the entire battery of original, multi-item validated scales, administered digitally within the AuxiliApp (see Figure 1). Each of these original scales preserved its specific, original response format (e.g., categories, numerical ranges, or yes/no options as per its design). Following the completion of all original scales, participants then proceeded to complete a collection of single slider-format items, each carefully selected by the research team as most representative for a specific construct (e.g., mood, loneliness, impulsivity). These slider items universally utilized a 0–10 continuous scale. All assessments were conducted sequentially within a single digital session on the AuxiliApp (https://psiquiatria.com/publico/index.php; accessed on 20 December 2024), ensuring a consistent testing environment for all participants. This sequential administration within a single session allowed for the assessment of convergent validity between the traditional multi-item scales and their novel slider-format counterparts, thereby evaluating the psychometric equivalence of these different assessment formats for the same constructs.

The development of the slider-format items for AuxiliApp was a meticulous process undertaken by the research team. The team comprises university professors and distinguished academics with extensive expertise in suicidology, psychometrics, and instrument development. Notably, several members, including Professors Sáiz Martínez, Pilar, Seoane Rey, Julio, Giner Ubago, and José (all full professors), have a proven track record in both developing novel psychological instruments (e.g., the Oviedo Sleep Questionnaire; [72] and rigorously adapting numerous internationally recognized scales into Spanish (e.g., Social Anxiety and Distress Scale, Sheehan Disability Inventory; [12]; MINI International Neuropsychiatric Interview; [28]; EuropASI; [11]; Clinician Administered PTSD Scale (CAPS); [33], among others).

The decision-making process for item selection and their subsequent adaptation to the slider format was achieved through multiple consensus meetings among the research team. These discussions leveraged the team’s collective clinical and psychometric expertise, informed by a thorough review of existing scientific literature on digital assessment and the psychometric properties of the original scales. The primary objective was to create digital equivalents that maintained the psychometric integrity and semantic content of established, validated instruments.

Specifically, the slider-format items were conceived as a direct, literal, one-to-one adaptation of the original items from the selected standardized scales. Minor linguistic adjustments were applied to enhance clarity and natural flow within a digital self-report context, while scrupulously preserving the semantic meaning of the original questions. For instance, an item like “I keep calm” from the original STAXI scale was rephrased to “¿Mantiene la calma?” (Do you keep calm?) or “I get angry easily” to “¿Se enfada con facilidad?” (Do you get angry easily?) in the slider format. These adaptations were solely aimed at optimizing readability and user experience within the app without altering the underlying construct being measured.

The rationale for reversing certain items was grounded in psychometric principles. For the majority of scales, a higher numerical score consistently indicates a greater severity or presence of a symptom. For items or subscales designed to measure the absence of a symptom or the presence of a protective factor (e.g., reasons for living), the response scale for the slider was intentionally reversed. This ensured that, for all measured dimensions, a higher numerical value assigned by the slider consistently reflected a greater level of the measured construct (e.g., more reasons for living), thereby maintaining a uniform interpretative direction for all data analysis.

### 2.5. Instruments

All instruments were administered digitally through AuxiliApp, presenting two distinct formats: the original validated Likert-type scale format and a novel slider-format adaptation. Participants completed both versions for each dimension. The dimensions assessed were suicidal intent, impulsivity, aggressiveness, anger, depression, reasons for living, loneliness, and psychological pain.

Beck Depression Inventory-II (BDI-II): Assesses the severity of depressive symptoms ([9]; Spanish adaptation by [81], which reported an excellent internal consistency of α = 0.89 in the general population). The scale consists of 21 items rated on a 4-point scale from 0 to 3, covering emotional, cognitive, and somatic aspects of depression, such as hopelessness, guilt, and physical discomfort.Psychache Scale: Measures psychological pain using 13 items rated on a 5-point Likert scale ([43]; Spanish adaptation by [65], which reported an excellent internal consistency of α = 0.90). Higher scores indicate greater levels of psychological distress.Paykel Suicide Scale (PSS): Assesses the severity and frequency of suicidal ideation. For this study, we utilized a Spanish adaptation for adolescents ([29]; see also [7], who reported an excellent internal consistency of α = 0.94 in a Peruvian adolescent sample).State-Trait Anger Expression Inventory-2 (STAXI-2): Evaluates the experience, expression, and control of anger ([85]; [58]). The scale comprises several subscales, including State Anger, Trait Anger, Anger Expression-Out, Anger Expression-In, Anger Control-Out, and Anger Control-In. Psychometric studies of its Spanish version have demonstrated good internal consistency, with Cronbach’s alpha values ranging from 0.70 to 0.90 for its various subscales (e.g., [30]).UCLA Loneliness Scale: Designed to assess perceived loneliness and social isolation. It contains 20 items rated on a 4-point scale, with higher scores indicating more frequent experiences of loneliness ([78]; [96]; see also [97]) has shown high reliability, with an excellent internal consistency of α = 0.90.Barratt Impulsiveness Scale (BIS): A widely used self-report measure of impulsivity ([71]). For this study, the Spanish adaptation for adolescents ([55]) was utilized, which has demonstrated good psychometric properties, including excellent internal consistency for the total scale (α = 0.89) and good consistency for its subscales (Attentional Impulsivity: α = 0.70; Non-planning Impulsivity: α = 0.81; Motor Impulsivity: α = 0.79).Reasons for Living Inventory (RFL): Measures an individual’s perceived reasons for not committing suicide ([51]). The Spanish adaptation ([64]) has shown excellent internal consistency, with a Cronbach’s alpha of α = 0.94 for the total scale, and values ranging from 0.70 to 0.89 for its various subscales (Survival and Coping Beliefs, Responsibility to Family, Child Concerns, Fear of Suicide, Fear of Social Disapproval, and Moral/Religious Objections).

These instruments were selected for their relevance to suicide risk and their established psychometric properties. Each was paired with a slider-format item designed to assess the same construct in a more concise and accessible visually intuitive format within the AuxiliApp.

### 2.6. Statistical Analysis

Pearson product–moment correlation coefficients were calculated to assess the convergent validity between scores obtained from traditional multi-item scales and their corresponding slider-format items for each relevant construct ([16]). This analysis aimed to evaluate the psychometric equivalence of the novel slider format against established measures.

Regarding the distribution of continuous variables, Pearson’s correlation coefficient was used due to its ability to measure the linear relationship between variables. Given the continuous nature of the data and the sample sizes for each correlation (ranging from N = 56 for the Beck Depression Inventory to N = 274 for the Reasons for Living Inventory), this statistic was considered appropriate for the purposes of this study, relying on the robustness of Pearson’s *r* in samples of this size.

## 3. Results

Prior to presenting the main analyses, it is important to address the varying sample sizes across the instruments. While the majority of the scales included in AuxiliApp were completed by over 200 participants, the Beck Depression Inventory-II (BDI-II) had a substantially smaller valid sample size of N = 56. This discrepancy was primarily attributed to participant dropout or incomplete responses, likely due to the BDI-II being administered towards the end of a comprehensive battery of questionnaires. Given the extended length of the overall assessment, it is plausible that some participants discontinued the survey before completing this final instrument. All subsequent analyses involving the BDI-II are based on this sample size, and this limitation is acknowledged.

All Pearson correlations between the traditional scale scores and their corresponding slider-format items were statistically significant (*p* < 0.001). According to Guilford’s criteria ([34]), most correlations were moderate (r = 0.40–0.70), while a subset were low (r = 0.20–0.40), particularly in subscales related to anger expression and family-based protective beliefs.

Moderate correlations were observed between the slider items and the traditional instruments assessing depression (BDI-II, r = −0.541), psychological pain (Psychache Scale, r = 0.598), suicidal ideation (Paykel, r = 0.503), loneliness (UCLA, r = 0.582), and several impulsivity and anger-related dimensions. Specifically, the BIS subscales showed correlations of r = −0.581 (Cognitive/Attentional), r = −0.445 (Non-Planning), and r = 0.426 (Motor). For anger, STAXI-2 subscales showed moderate correlations in State Anger (r = 0.459), Trait Anger (r = 0.558), and Anger Expression-In (r = 0.489).

Lower correlations were found for STAXI-2 subscales of Anger Expression-Out (r = 0.292), External Anger Control (r = 0.354), and Internal Anger Control (r = 0.380). Similarly, within the Reasons for Living Inventory (RFL), the Survival and Coping Beliefs subscale showed a moderate negative correlation (r = −0.589), while the Family Responsibility and Child-Related Concerns (r = −0.382), Fear of Suicide (r = −0.283), Fear of Social Disapproval (r = −0.455), and Moral Objections (r = −0.359) subscales showed low correlations.

In all inversely worded items, higher traditional scale scores (indicating greater symptomatology or protective beliefs) corresponded to lower scores on the associated slider question, accounting for the negative correlations. Overall, the data supports the validity of slider-format questions as a simplified yet valid alternative for assessing core dimensions related to suicide risk.

A detailed comparison of the means and correlations between the traditional and slider-format items is presented in Table 1.

## 4. Discussion

Our findings indicate that slider-format items can approximate scores from validated instruments with moderate correlations in most domains, including depression, psychological pain, suicidal ideation, loneliness, and core aspects of impulsivity and anger. Specifically, these moderate correlations provide evidence of convergent validity for the slider items across these dimensions. The demonstrated feasibility of digitally assessing constructs like impulsivity and anger, consistently identified as key risk factors for suicidal behavior ([59]; [13]; [48]), underscores the potential of mobile applications to facilitate timely and efficient risk assessments. While lower correlations were observed in some subscales—particularly those assessing nuanced components of anger regulation and certain protective beliefs against suicide—this does not diminish the potential utility of digital sliders, especially considering their brevity, accessibility, and ease of implementation.

The broader relevance of telemedicine has become undeniable, especially during the COVID-19 crisis. The pandemic not only challenged the capacity of healthcare systems, with a reported 90% of countries experiencing disruptions in essential health services ([101]), but also profoundly reshaped how care is delivered and received. Many individuals refrained from seeking care for non-COVID-related issues ([60]), negatively affecting public health outcomes. The pandemic’s psychological impact is well documented: there has been a marked increase in the prevalence of mental disorders ([80]; [100]), worsened outcomes for individuals with pre-existing psychiatric conditions ([61]), and heightened suicide risk ([23]; [45]). Some have even described this as a parallel pandemic of mental health ([70]) or predicted that mental illness will represent the next global health crisis ([66]).

In this context, the integration of new technologies into clinical practice has transitioned from a complementary approach to a necessary evolution in care delivery. International healthcare systems rapidly pivoted from in-person consultations to virtual modalities ([98]), reinforcing the need for tools that are compatible with remote assessment. Accurate and timely identification of individuals at risk is paramount for effective prevention strategies. Traditional methods of psychological assessment, though well-established, can sometimes be lengthy, resource-intensive, or less engaging, particularly for younger, digitally native populations ([54]).

The present study sought to explore whether simplified, slider-format questions administered via a mobile app could serve as psychometrically equivalent and valid alternatives to traditional psychometric instruments in assessing suicide-related dimensions. The existing literature largely supports the digitalization of self-report measures. Rutherford’s meta-analysis ([79]) concluded that paper-based and electronic formats can be used interchangeably under appropriate conditions. Other researchers, while more cautious, recognize the advantages of digital adaptation, though they advise against assuming full psychometric equivalence without prior validation ([3]; [8]; [35]).

Slider questions provide a practical and highly intuitive format for widespread use across digital platforms, such as mobile phones and tablets. Their continuous visual nature offers a more nuanced response scale than discrete options, eliminating explicit numerical choices and facilitating a direct drag-and-drop interaction common in modern digital interfaces ([75]).

These features, combined with their inherent brevity, enhance user engagement and allow for broader reach, especially in underserved or geographically isolated populations ([56]). Unlike lengthy standardized scales, slider items can be completed quickly and easily, offering a feasible solution for both large-scale screening and ongoing patient monitoring.

Furthermore, the integration of digital tools, such as mobile applications, into mental healthcare delivery has gained significant traction, particularly in the context of suicide prevention. Evidence suggests that web-based and mobile interventions can play a vital role in various aspects of suicide prevention, including delivering psychoeducation, providing crisis support, enhancing coping skills, facilitating self-monitoring of mood and suicidal ideation, and improving access to care for underserved populations ([42]; [50]; [63]; [4]). For instance, apps offering immediate access to helplines or safety planning tools have shown promise in acute crisis situations, while those incorporating cognitive behavioral therapy (CBT) or dialectical behavior therapy (DBT) elements can foster long-term resilience and skill development ([83]). While promising, a systematic review of mobile applications for depression and suicide prevention revealed that only a small fraction incorporated evidence-based strategies to address suicidal behavior ([91]), highlighting a critical gap. Our study addresses this by specifically validating a novel, user-friendly assessment format that can enhance the utility and applicability of these technologies in mental health assessment and proactive suicide prevention.

### Future Directions

Beyond the current findings, future research should explore more advanced psychometric properties and clinical applications of slider-based assessments. Specifically, assessing whether the psychometric equivalence and reliability of these scales vary among individuals with differing levels of suicidal ideation would be a highly valuable next step. Such subgroup analyses could reveal important nuances in the performance of these digital tools across different risk profiles, further enhancing their precision and clinical utility in targeted suicide prevention efforts. Additionally, longitudinal studies are needed to evaluate the test–retest reliability of slider items over time and their sensitivity to change following interventions, providing further evidence for their dynamic assessment capabilities. The feasibility of integrating such tools into clinical workflows warrants further exploration, particularly in primary care, university health services, and other first-line support environments.

## 5. Conclusions

In conclusion, this study provides compelling evidence for the psychometric equivalence and utility of slider-format questions integrated into a mobile app for assessing key dimensions related to suicide risk. This novel approach offers a highly intuitive and engaging method for data collection, potentially enhancing user compliance and data quality in self-report assessments. While these simplified tools should not replace comprehensive clinical evaluations, they hold promise for broad applications in early screening, symptom monitoring, and follow-up in mental health contexts. Their digital format aligns well with contemporary telemedicine practices and may help overcome structural and social barriers to care. The primary contribution of this research lies in validating a unique, user-friendly digital assessment format that goes beyond traditional Likert scales, offering a continuous and visually interactive response mechanism. This innovation has significant implications for advancing the field of digital mental health, particularly in suicide prevention, by facilitating rapid, sensitive, and accessible screening and monitoring of risk factors. Our findings support the integration of such innovative digital tools into comprehensive suicide prevention strategies, paving the way for more dynamic and personalized interventions. Continued research is needed to evaluate the clinical utility and diagnostic performance of slider-based assessments across diverse populations and settings.

## 6. Strengths and Limitations

One of the key strengths of this study lies in its innovative use of slider-format items within a mobile application to assess complex psychological constructs associated with suicide risk. The digital nature of the tool offers advantages in terms of cost-efficiency, accessibility, and scalability, making it well-suited for large-scale implementation and routine screening. Moreover, the inclusion of a young university population—typically comfortable with digital platforms—enhances the ecological validity of the method. A key novelty of this study lies in its comprehensive validation of a broad range of suicide-risk factors using the innovative slider format delivered through a dedicated mental health application (AuxiliApp).

However, this study is not without limitations. First, the sample consisted exclusively of university students, limiting the generalizability of the findings to broader or clinical populations. While this population is highly relevant given the rising rates of mental health concerns and suicide risk in young adults, it is also characterized by a higher level of education and digital literacy, which may have contributed to the observed high usability and acceptance of the mobile application. Consequently, the generalizability of our findings to broader or more diverse populations, including individuals with lower educational attainment, older adults, or those with limited technological proficiency, cannot be assumed. Future research should aim to replicate these findings in more heterogeneous populations to establish the broader applicability of slider-based assessment formats in diverse clinical and general settings. Second, while correlations between slider and traditional items were statistically significant, some remained in the low range, particularly in constructs related to anger control and moral or familial reasons for living. Additionally, the study did not assess predictive validity or compare diagnostic performance, nor did it establish specific clinical cutoffs or diagnostic thresholds for the novel slider format. The current 0–10 scale for slider items, while intuitive, lacks the direct interpretative or diagnostic capability of traditional scales with established severity categories. Finally, inverse wording in some slider items may have introduced interpretive variability.

## Figures and Tables

**Figure 1 behavsci-15-00868-f001:**
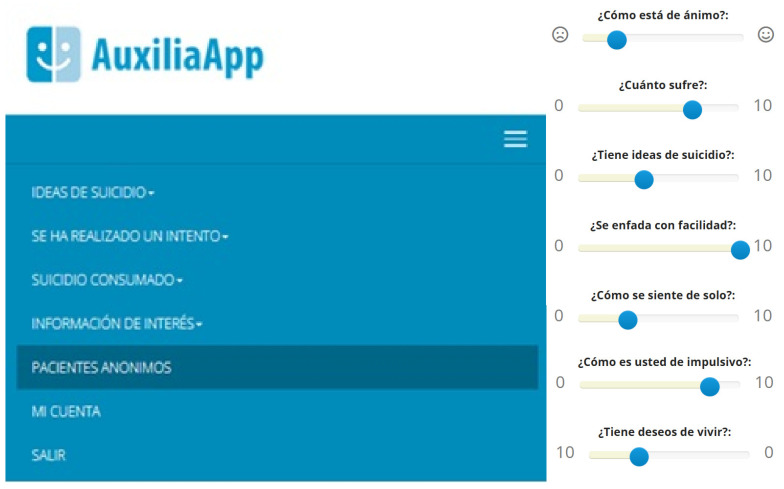
Main menu of the AuxiliApp web version.

**Table 1 behavsci-15-00868-t001:** Summary of results from traditional format and electronic slider format.

Scale	Slider Question	N	Traditional Format Mean (Min–Max)	Slider Format Mean (Min–Max)	Pearson’s r	*p*-Value
* Beck Depression Inventory	How is your mood?	56	5.268 (0–32)	7.197 (0–10)	−0.541	<0.001
Psychache Scale	How much are you suffering?	265	22.694 (13–64)	2.385 (0–9)	0.598	<0.001
Paykel	Do you have suicidal thoughts?	223	0.973 (0–5)	0.883 (0–10)	0.503	<0.001
STAXI-2—State Anger	Do you feel angry?	270	18.622 (15–50)	1.541 (0–8)	0.459	<0.001
STAXI-2—Trait Anger	Do you get angry easily?	270	21.152 (10–37)	4.037 (0–10)	0.558	<0.001
STAXI-2—Anger Expression-Out	Do you express your anger?	270	12.296 (6–23)	4.852 (0–10)	0.292	<0.001
STAXI-2—Anger Expression-In	Do you keep your anger to yourself?	270	12.456 (6–22)	3.889 (0–10)	0.489	<0.001
STAXI-2—External Anger Control	Do you remain calm?	270	17.900 (6–24)	5.756 (0–10)	0.354	<0.001
STAXI-2—Internal Anger Control	Do you count to ten before expressing your anger?	270	14.463 (6–24)	3.118 (0–10)	0.380	<0.001
UCLA Loneliness Scale	How lonely do you feel?	249	12.076 (0–58)	2.531 (0–10)	0.582	<0.001
* BIS—Cognitive/Attentional	Do you find it easy to concentrate?	272	12.658 (2–28)	6.224 (0–10)	−0.581	<0.001
* BIS—Non-Planning	Do you finish what you start?	272	14.158 (2–40)	7.404 (0–10)	−0.445	<0.001
BIS—Motor	How impulsive are you?	272	14.856 (2–39)	5.353 (0–10)	0.426	<0.001
* RFL—Survival and Coping	Do you have a desire to live?	274	19.467 (0–118)	9.310 (0–10)	−0.589	<0.001
* RFL—Family-Related	Does your family motivate you to live?	274	11.912 (0–50)	9.164 (0–10)	−0.382	<0.001
* RFL—Fear of Suicide	Are you afraid of suicide?	274	22.580 (0–35)	5.967 (0–10)	−0.283	<0.001
* RFL—Fear of Disapproval	Are you concerned about what people would think if you died by suicide?	274	10.117 (0–15)	3.817 (0–10)	−0.455	<0.001
RFL—Moral Objections	Do you think suicide is wrong?	274	13.496 (0–20)	7.256 (0–10)	−0.359	<0.001

Note: * The slider question is inversely worded. N = number of participants with complete data for each variable. BDI-II: Beck Depression Inventory-II, Paykel: Paykel Suicidal Ideation Scale, STAXI-2: State-Trait Anger Expression Inventory-2, UCLA: University of California, Los Angeles, BIS: Barratt Impulsiveness Scale, RFL: Reasons For Living Inventory, Min-Max: Minimum-Maximum.

## Data Availability

The data supporting the findings of this study are available from the corresponding author upon reasonable request.

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
