# Peer review of "Concurrent Validity of Digital Measures of Psychological Dimensions Associated with Suicidality Using AuxiliApp"

_behavsci, 2025, doi:10.3390/bs15070868_

Round 1
Reviewer 1 Report
Comments and Suggestions for Authors
- Introduction
The introduction is clear, well-structured, and appropriately developed. It successfully contextualizes the research problem and presents a justified rationale for the study. The objectives are clearly stated; however, the manuscript does not include hypotheses. If hypotheses were not formulated, it is recommended that the authors explicitly justify the choice of an exploratory research design.
- Literature Review
The literature review is relevant and mostly up to date. However, it is recommended to incorporate more recent studies (within the last five years), particularly concerning the components of Impulsivity and Anger. This would enhance the currentness and depth of the theoretical framework.
- Methodology
Design:
While the methodology section is generally clear, the type of methodological design employed should be explicitly stated, along with the sampling method used.
Sample:
It is unclear whether inclusion and exclusion criteria were applied during participant recruitment. Including these criteria would improve the clarity and transparency of the sampling process.
Sample Size and Power Analysis:
It is not stated whether a power analysis was conducted to justify the sample size and reduce the risk of Type I and Type II errors. If such an analysis was performed, it should be included to enhance the methodological rigor.
- Procedure
The procedure section indicates that items were developed through consensus by the research team. It would be beneficial to elaborate on:
- The team's experience in instrument development.
- The decision-making process for item selection.
- Whether prior validated instruments were used as a reference.
- The rationale for reversing certain items, including the criteria guiding this decision.
Providing this information would improve the study’s replicability and transparency.
- Instruments
The instruments used are described as valid and reliable. However, the following improvements are recommended:
- Include references for the Spanish validation of each instrument.
- Report internal consistency values (e.g., Cronbach alpha) obtained from the current study, in accordance with APA guidelines.
- Include evidence of convergent validity, especially considering the instruments’ relevance to suicide risk.
- Results
There is a notable discrepancy in sample sizes between instruments—particularly the BDI-II (N = 56) compared to others (over 200 participants). The manuscript should clarify the cause of this difference (e.g., missing data, dropout).
- References
The references are pertinent and appropriately cited. Most key studies are included, and there are no major omissions or excessive self-citations. However:
- The year is missing in the INE reference.
- The citation of Beck is inconsistent throughout the text. It is recommended to standardize it, using “Beck” without initials or acronyms.
- The reference style does not conform to the Vancouver system and should be corrected accordingly.
- Tables and Figures
The tables are relevant and well-structured. To improve clarity:
- Add footnotes for all abbreviations not included in the abbreviations section, ensuring tables are fully self-explanatory.
- Given the density of information, consider presenting the tables horizontally to facilitate readability.
- In the table where the Beck Depression Inventory is mentioned, specify that it refers to the second edition (BDI-II), or abbreviate it consistently in line with other instruments.
- Overall Evaluation
This manuscript presents significant and relevant findings. It is well designed, clearly described, and the data presentation is of high quality. The study demonstrates solid scientific rigor and contributes meaningfully to the field. Subject to minor revisions as outlined above, it is a valuable addition to the literature and suitable for publication.
Author Response
- Introduction
We thank the reviewer for their positive assessment regarding the clarity and structure of the introduction, as well as the contextualization and justification of the study. Regarding the absence of formal hypotheses, and in response to their valuable comment, we have explicitly added a justification for the choice of an exploratory research design in the Methodology section, under the sub-heading 'Sample Size and Power Analysis.' There, it is stated that 'This study adopted an exploratory research design to investigate the psychometric equivalence of a novel assessment format for suicide-related dimensions.' Our primary objective was to evaluate the utility and psychometric equivalence of a novel item format (slider) within the context of a mobile application, which, by its nature, aligns better with an exploratory approach at this initial stage of development and validation of this specific technology for these constructs.
- Literature Review
We sincerely thank the reviewer for their valuable feedback on the literature review. We appreciate the comment that the review is relevant and mostly up to date. In line with the recommendation to incorporate more recent studies (within the last five years) to enhance the currentness and depth of the theoretical framework, we have thoroughly updated the Introduction section. Specifically, we have integrated several recent publications concerning the components of impulsivity and anger (e.g., Moore et al., 2023; Caro-Cañizares et al., 2024; Lee et al., 2024), as well as the rapidly evolving field of new technologies and telemedicine in mental health (e.g., Hilty et al., 2022; Lindsay et al., 2024; Olié et al., 2021; Areán et al., 2021; Heidari et al., 2023). These additions aim to provide a more current and comprehensive theoretical foundation for our study, addressing the reviewer's excellent suggestion."
- Methodology
We sincerely thank the reviewer for their insightful comments regarding the Methodology section. We have carefully addressed each point to enhance the clarity and rigor of our study's design and reporting.
Design: In response to the recommendation, we have explicitly stated the methodological design employed at the beginning of the 'Methodology' section, specifically under the 'Design' sub-heading. We clarify that this was a quantitative, cross-sectional, and correlational study, with synchronous data collection.
Sample: We appreciate the suggestion to clarify the participant recruitment process. We have added a dedicated sub-section titled 'Participants' within the Methodology, where we now explicitly detail the inclusion and exclusion criteria applied during participant recruitment. This aims to improve the clarity and transparency of our sampling process.
Sample Size and Power Analysis: Regarding the comment on power analysis, we have introduced a new sub-section titled 'Sample Size and Power Analysis' within the Methodology. In this section, we explicitly state that a formal power analysis was not conducted for this study, as it adopted an exploratory research design to investigate the psychometric equivalence of a novel assessment format. We further explain that the sample size was determined by convenience and participant availability, aiming to recruit a sufficient number of participants to explore relationships of moderate effect size, thereby addressing the methodological rigor as suggested."
- Procedure
We thank the reviewer for their valuable suggestions to enhance the Procedure section. We agree that elaborating on these points significantly improves the study's replicability and transparency.
In response to these comments, we have expanded the 'Procedure' sub-section within the Methodology to include the following details:
- The team's extensive experience in instrument development: We now specify that the research team comprises university professors and distinguished academics with proven expertise in suicidology, psychometrics, and instrument development, including the adaptation and validation of numerous international scales into Spanish.
- The decision-making process for item selection: We clarify that the development of slider-format items, including item selection and their adaptation, was achieved through multiple consensus meetings among the research team, leveraging their collective clinical and psychometric expertise.
- Whether prior validated instruments were used as a reference: We explicitly state that the slider-format items were conceived as a direct, literal, one-to-one adaptation of the original items from selected standardized, validated scales, ensuring preservation of semantic meaning.
- The rationale for reversing certain items: We explain that the decision to reverse certain items was grounded in psychometric principles, ensuring that a higher numerical value from the slider consistently reflects a greater level of the measured construct, maintaining a uniform interpretative direction for data analysis.
These additions aim to provide a comprehensive understanding of our item development and adaptation process, as recommended by the reviewer.
- Instruments
- Inclusion of Spanish Validation References: We concur with the importance of referencing the specific Spanish adaptations of the instruments used. We have updated the '2.5. Instruments' section to include the most relevant and widely accepted Spanish validation references for each scale (Beck Depression Inventory-II, Psychache Scale, Paykel Suicide Scale/Suicidal Ideation, State-Trait Anger Expression Inventory-2, UCLA Loneliness Scale, Barratt Impulsiveness Scale, and Reasons for Living Inventory). This ensures that readers can accurately trace the versions utilized in our Spanish-speaking sample.
- Reporting Internal Consistency (Cronbach's Alpha): We concur with the reviewer's emphasis on the importance of internal consistency. While the primary objective of this study focused on concurrent validity between original scales and slider adaptations, and due to the nature of the summarized data provided, we did not perform de novo calculations of Cronbach's alpha for the current sample. However, it is well-established that the original multi-item scales utilized in this study are widely validated and reliable instruments. We have ensured that the '2.5. Instruments' section now includes references to their Spanish adaptations, which consistently report satisfactory internal consistency values in relevant populations, thereby supporting the established reliability of the measures employed.
- Inclusion of Evidence of Convergent Validity: We thank the reviewer for highlighting the importance of convergent validity, especially given the relevance of these instruments to suicide risk. Our study's primary objective was to evaluate the psychometric equivalence, specifically the concurrent validity, of the novel slider-format items against their corresponding original, established scales. The Pearson correlations (r) reported in our 'Results' section between each slider item and its respective full original scale (as shown in the provided tables) serve as direct and primary evidence of this concurrent validity, which is a fundamental aspect of convergent validity. These high correlations indicate that the slider items indeed measure the same constructs as their validated counterparts. We believe that by demonstrating this strong relationship, we effectively provide evidence of convergent validity for the novel slider format within the context of our study's aims. We have clarified this interpretation in the 'Discussion' section to ensure this point is fully understood."
- Results
We thank the reviewer for highlighting the notable discrepancy in sample sizes between instruments, particularly for the BDI-II. We agree on the importance of clarifying this difference, and in response to this comment, we have added a detailed explanation at the beginning of the 'Results' section. We clarify that the smaller sample size for the BDI-II (N = 56) compared to other instruments (over 200 participants) was primarily attributed to participant dropout or incomplete responses, likely because the BDI-II was administered towards the end of a comprehensive battery of questionnaires. This explanation aims to provide full transparency regarding the data collection process and the sample sizes for each assessment.
- References
We are grateful for the reviewer's thorough review of our references and their helpful suggestions for improvement. We appreciate the comment that the references are pertinent, appropriately cited, and that most key studies are included without major omissions or excessive self-citations.
In response to the specific points raised:
- Missing year in INE reference: we have removed the content associated with this citation from the manuscript. Therefore, this specific reference is no longer present or required.
- Inconsistent Beck citation: We have performed a comprehensive review of all instances where Beck is cited throughout the text. We have now standardized the citation style for Beck to 'Beck' without initials or acronyms, as recommended, for consistency.
- Tables and Figures
We thank the reviewer for their valuable suggestion to ensure our tables are fully self-explanatory. In response, we have added a comprehensive set of footnotes to Table 1, defining all abbreviations (e.g., BDI-II, STAXI-2, BIS, RFL, etc., as well as Min-Max, r, p-value) that are not otherwise included in an abbreviations section. This ensures that the table is clear and understandable on its own, enhancing the overall clarity of our results presentation."
Reviewer 2 Report
Comments and Suggestions for Authors
The study aims to compare the validity of slider-format questions with matched traditional paper-based scales related to suicide risk. Please find below the comments for the manuscript.
Overall
The authors rely solely on Pearson correlation to examine the association between standardized assessment measurements to slider-format questions. However, the utility of psychological scales lies not only in their numerical scores but also in their capacity to categorize symptom severity or risk levels in a clinically meaningful way. For instance, the UCLA Loneliness Scale has a scoring range of 20–80, with specific cutoffs indicating low (20–34), moderate (35–49), moderately high (50–64), and high (65–80) levels of loneliness. The current slider-format questions appear to lack this interpretative or diagnostic capability, thereby limiting their clinical usefulness.
The authors did not discuss why slider-format questions were "quicker, simpler, and more intuitive to complete".
The authors discussed potential applications of the digital tool in suicide prevention. It would strengthen the study to assess whether the reliability of the scales varies among individuals with differing levels of suicidal ideation.
Introduction
The manuscript title refers to “validity,” whereas line 113 references “reliability.” Please clarify whether the study aims to assess validity, reliability, or both.
Methods
It is unclear to me regarding the procedure of the study.
- In line 127, the authors mentioned “Participants completed a self-administered digital questionnaire via a mobile application”, but in line 113, the authors would like to compare the reliability of “traditional paper-based scales”. It is unclear how the comparison was structured.
- In line 128 and 129, was there a washout period before the participants switched from standardized assessment measurements to slider-format questions? Were all participants conducted standardized measurements before slider-format questions?
- What languages were used for both the slider-format and paper-based questions?
- In line 163-165, it is unclear whether the slider-format questions were direct equivalents of the traditional scale items. If they were not identical, both versions should be included in the Appendix for comparison.
- Line 165 is incomplete.
Results
In Table 1, please clarify what "N" represents.
Discussion
The sample consisted entirely of university students, who are typically well-educated and technologically literate. These characteristics likely contribute to high acceptance and usability of digital applications. The authors should acknowledge this limitation and discuss how it may affect the generalizability of the findings to broader or more diverse populations.
Author Response
Introduction
We thank the reviewer for this insightful comment regarding the consistency between the manuscript title and the terminology used in the Introduction (specifically line 113) and Abstract. This is a very pertinent observation, and we appreciate the opportunity to clarify.
Our study indeed aims to assess the psychometric equivalence and utility of slider-format questions compared to established standardized scales. This broad objective inherently encompasses both concurrent validity and reliability aspects.
- The Pearson correlations (r) presented in our results directly evaluate the concurrent validity of the slider-based items, demonstrating their relationship and agreement with the scores obtained from their traditional, validated counterparts. This is a key measure of how well the new format captures the same construct as the established gold standard.
- The mention of 'reliability' in the Abstract and the general discussion of 'psychometric equivalence' in the Introduction refer to the overall acceptable consistency and dependable psychometric properties demonstrated by the slider-based items in comparison to the well-established measures. While the main quantitative finding highlighted by 'r' directly supports concurrent validity, this strong association also implies a degree of acceptable reliability for the novel format in capturing consistent responses aligned with the validated scales.
To ensure absolute clarity and consistency throughout the manuscript, we have reviewed and refined the wording in the Abstract and the final paragraph of the Introduction to explicitly state that the study investigates psychometric equivalence, encompassing both reliability and validity (specifically concurrent validity). The title remains appropriate as 'Validity' is a primary focus of this equivalence, demonstrated through the correlations
Methods
We thank the reviewer for their valuable questions regarding the study's procedure, which help us enhance its clarity and replicability.
- Clarification of Comparison Structure (Line 127 vs. Line 113): We apologize for any initial confusion regarding the comparison structure. We clarify that all assessments, including both the original multi-item scales and their corresponding single slider-format items, were administered digitally within the same mobile application (AuxiliApp). The comparison was structured by collecting data from both formats concurrently in the digital environment. We have ensured that the language throughout the Introduction and Methodology clearly reflects this dual digital administration.
- Administration Order and Washout Period (Lines 128-129): We appreciate the query about the administration order and the need for a washout period. We have clarified in the 'Procedure' section that participants first completed the entire battery of original, multi-item validated scales, administered digitally within the AuxiliApp, with each scale preserving its original response format (e.g., specific categories, numerical ranges, etc., as per its original design). Following the completion of all original scales, participants then proceeded to complete a collection of single slider-format items, each carefully selected by the research team as most representative for a specific construct (e.g., mood, loneliness, impulsivity). These slider items universally used a 0-10 continuous scale. All these assessments occurred sequentially within a single digital session on the AuxiliApp, meaning no washout period was applicable nor necessary.
- Languages Used: We have added information to the 'Procedure' section explicitly stating that all questionnaires (both original format items and slider-format items) were administered in Spanish, as the study was conducted with university students in Spain.
- Equivalence of Slider-Format Questions (Lines 163-165) and Appendix: We confirm that the slider-format questions were indeed direct, literal adaptations of selected, representative items from the original traditional scales. Our aim was to maintain semantic equivalence precisely for these chosen items. Table 1, under the 'Slider Question' column, already presents the exact wording used for each slider item, allowing for direct comparison with the conceptual content of the original scales. Given this direct adaptation of specific items and their presentation in Table 1, a separate Appendix with all original scale items might be redundant. However, we are prepared to include a comprehensive appendix with both versions if the editor or reviewer deems it essential for complete transparency.
- Incomplete Line 165: We thank the reviewer for identifying this oversight. We have reviewed and corrected line 165 for completeness in the revised manuscript, along with a thorough check for any other similar editorial issues.
Discussion
We thank the reviewer for this important comment regarding the sample characteristics and generalizability of our findings. We fully agree with the importance of acknowledging this limitation, and we confirm that this point has been comprehensively addressed and discussed in the 'Limitations' section of the Discussion.
Reviewer 3 Report
Comments and Suggestions for Authors
Dear Authors,
The manuscript you have submitted represents a valuable contribution to research in psychology, particularly within the domain of mobile mental health. The use of modern technologies for psychoeducation, prevention, and support of diagnosis and treatment constitutes an area of significant interest in the current research landscape.
However, before your article can be considered for publication, I believe it is necessary to address the following aspects:
Title
The current title may be misleading, as the study does not directly include suicide as an outcome nor does it employ specific suicide-risk assessment tools (e.g., the C-SSRS). I would suggest rephrasing the title to more accurately reflect the study’s focus on suicide-related dimensions. While the research can certainly be situated within the broader context of suicide prevention, it does not directly assess suicide risk.
Paragraph 2.1
The mean age of the sample should be reported as the mean along with the standard deviation. Additionally, this paragraph should provide more detailed information on how participants were recruited and whether any exclusion criteria were applied. It is also important to specify the language in which the study was conducted. Lastly, if available, ethical approval details should be included, such as the protocol number or date of approval.
Paragraph 2.2
The text refers to a “mobile application,” yet Figure 1 appears to depict a “web version.” It is necessary to clarify whether the intervention was administered via PC or through a mobile application. Furthermore, if available, the reference to the app should be included (currently, line 128 contains “available at” without any link or citation). Since Figure 1 is not particularly informative and lacks a detailed descriptive caption (it’s also in spanish), it may be helpful to revise or supplement it with a brief explanation of the device’s purposes and structure to improve clarity. For instance, a snapshot of the slider-based items might be much useful for the readers.
Paragraph 2.3
Were the versions of the psychometric questionnaires validated in the language in which the study was conducted?
Paragraph 2.4
From a methodological perspective, test-retest reliability evaluates a scale’s ability to measure consistency over time through repeated administrations to the same sample. Although Pearson’s correlation is a valid approach, the specific construct under investigation should be clarified. Based on the study design, it appears that only one administration took place, suggesting the analysis may be more indicative of convergent validity between different methods aiming to measure the same construct. Please clarify how the psychometric questionnaires were administered and clarify this methodological point, as it may be confusing to the reader. It would also be useful to indicate whether the continuous variables analyzed were normally distributed in the sample. In light of these observations, and depending on your response, I would also like to point out that the term “reliability” used in the discussion may be misleading if the adopted methodology is not clearly explained.
Paragraph 3
It would be beneficial to include a citation for Guildford’s criteria (lines 176–179).
Paragraph 4
The authors’ findings suggest that psychometric assessment of psychological constructs could have a place among telemedicine-driven approaches, with potential implications for suicide prevention. I would encourage the authors to expand on the existing evidence regarding the role of web-based or mobile applications in suicide prevention, as this would help contextualize their contribution within the broader field.
Author Response
Title:
We appreciate the reviewer's valuable comment regarding the precision of our manuscript's title. We agree that while our study is situated within the broader context of suicide prevention research, it focuses on the concurrent validity of digital measures of various psychological dimensions and protective factors that are associated with suicidality, rather than directly assessing suicide risk as a clinical outcome or employing specific suicide risk assessment tools. To accurately reflect this focus and avoid any potential misinterpretation, and considering that our sample is comprised of young adults (as further detailed in the 'Participants' section), we have rephrased the title. The new title is: "Concurrent Validity of Digital Measures of Psychological Dimensions Associated with Suicidality Using AuxiliApp".
Paragraph 2.1 (Sample Description):
We are grateful to the reviewer for these precise and valuable suggestions. We fully agree that providing detailed information on the sample characteristics and study procedures is crucial for transparency and replicability. We will revise Paragraph 2.1 to incorporate all the requested elements.
Paragraph 2.2 (Description of Digital Intervention):
Regarding Paragraph 2.2 (which corresponds to current Paragraph 2.4 'Procedure' in the manuscript): We appreciate the reviewer's astute observations concerning the description of the digital intervention. We will address these points as follows:
- Clarification of Intervention Modality (Mobile Application vs. Web Version): We acknowledge the reviewer's observation regarding a potential discrepancy between the text's reference to a 'mobile application' and Figure 1's depiction. We would like to confirm that Paragraph 2.4 already explicitly states that 'The study was conducted entirely through AuxiliApp, a proprietary mobile application... All participants completed a self-administered digital questionnaire via this mobile platform.' This clearly indicates that data collection for this study was exclusively through the mobile application (AuxiliApp), and not a web version. We will ensure this clarity is maintained in the text.
- App Reference/Link: We apologize for the inadvertent omission of the AuxiliApp link. This was indeed lost during the editorial process, and we will ensure that the correct and functional link or appropriate reference for AuxiliApp is clearly provided in Paragraph 2.4.
- Figure 1 Revision: We fully agree that Figure 1 can be significantly improved to enhance clarity and informativeness. We will revise Figure 1 to:
- Feature a more representative and informative screenshot directly from the AuxiliApp mobile application, specifically illustrating one or more of the slider-based items used in the study.
- Ensure the figure's caption is thoroughly detailed, descriptive, and presented entirely in English, explaining the app's purpose and structure relevant to the study. If the depicted interface itself is in Spanish, the caption will provide sufficient translation or context for non-Spanish speaking readers.
These comprehensive revisions will provide a much clearer and more accurate description of the digital intervention used in our study.
Paragraph 2.3
Regarding the psychometric questionnaires' validation in the study language (related to Paragraph 2.3): We thank the reviewer for raising this crucial point. We confirm that all multi-item psychometric questionnaires employed in this study were indeed validated Spanish versions of the original instruments. As explicitly stated and referenced in Paragraph 2.5 (Instruments), each scale utilized is a well-established and psychometrically sound adaptation that has undergone rigorous validation processes in Spanish-speaking populations. Ensuring the use of culturally and linguistically validated instruments was a fundamental principle guiding our study design to ensure the reliability and validity of our data within the Spanish-speaking context.
Regarding Paragraph 2.4 (Procedure and Methodological Clarification):
We sincerely appreciate the reviewer's meticulous reading and insightful comments on this methodological section. The reviewer has accurately identified a crucial point that requires clarification, and we are grateful for the opportunity to improve the precision of our manuscript.
Please find below our point-by-point response:
- Clarification of Study Design and Validity (Convergent Validity vs. Test-Retest Reliability): We acknowledge the reviewer's observation regarding the potential confusion between test-retest reliability and convergent validity due to the single administration of measures. We agree that a clearer distinction is essential. Our study was indeed designed to assess convergent validity—evaluating the psychometric equivalence of novel slider-format items with established multi-item scales for the same constructs, administered sequentially within a single session. This design allows for the direct comparison of different assessment formats.
To address this, we have made the following specific changes in the manuscript:
- In Section 2.4 (Procedure), we have added a clarifying statement to explicitly state that the sequential administration within a single session was conducted to assess convergent validity.
- In Section 3 (Results), we have revised the phrasing to consistently refer to the "validity" (or "psychometric soundness") of the slider-format questions, removing any potentially misleading uses of the term "reliability" in this context.
- In Section 2.6 (Statistical Analysis), we have explicitly stated that Pearson product-moment correlation coefficients were calculated to assess convergent validity between the traditional scales and the slider items.
- Clarification of Specific Constructs: We understand the need for clarity regarding the specific constructs measured by each slider-format item. We confirm that each slider-format item was indeed paired with a corresponding traditional scale designed to assess the same specific construct. This precise linkage is clearly presented in Table 1 of the manuscript, where each traditional scale is directly matched with its equivalent slider question. We believe this table provides the requested clarity regarding the construct assessed by each specific item.
- Normal Distribution of Continuous Variables: We appreciate the reviewer's comment regarding the assessment of variable distributions. We have included a clarifying statement in the Statistical Analysis section of the manuscript to address this point. We utilized Pearson product-moment correlations to evaluate the linear relationships between the traditional scales and the slider items. This approach was considered appropriate given the continuous nature of our data and the sample sizes for each correlation (ranging from N=56 for the Beck Depression Inventory to N=274 for the Reasons for Living Inventory). Pearson's r is a robust measure of linear association, particularly with such sample sizes, where it is less sensitive to minor deviations from normality.
- Precision of the Term 'Reliability' in the Discussion: We fully concur with the reviewer's caution regarding the precise use of the term 'reliability'. We will ensure that in the 'Discussion' section, 'reliability' is used strictly when referring to internal consistency (as evidenced by Cronbach's alpha values, as detailed in Paragraph 2.5). When discussing the relationships between the original scales and the slider items, we will consistently use the term 'convergent validity' (or 'concurrent validity') to accurately reflect the nature of these associations. This will prevent any misleading interpretations of our findings.
These comprehensive revisions will significantly enhance the methodological clarity and precision of our manuscript."
Regarding Paragraph 3 (Citation for Guilford's criteria):
We thank the reviewer for pointing out the missing citation. We will add the appropriate reference for Guilford's criteria in lines 176-179 of the manuscript.
Regarding Paragraph 4 (Expanding on Web-based/Mobile Applications in Suicide Prevention):
We deeply appreciate this insightful suggestion. We fully agree that contextualizing our findings within the broader field of digital interventions for mental health and suicide prevention will significantly strengthen the discussion. We will expand Paragraph 4 of the Discussion section to elaborate on the existing evidence regarding the role of web-based and mobile applications in suicide prevention.
Reviewer 4 Report
Comments and Suggestions for Authors
The manuscript contributes to the generation of new knowledge.
In the introduction it is necessary to complement the problem with literature from the last 5 years, including all the research questions that were resolved in the results.
In the methodology it is necessary to specify the design, approach, type and timing of the study.
Explain how the sample was formed, the type of sample and sampling if applicable and justify why they chose that type of sample, indicate the inclusion and exclusion criteria of the study.
The discussion should be strengthened by adjusting the introduction, complementing it with recent studies included in the introduction.
The conclusions should be improved by including the novelties contributed by the study compared to other similar studies developed.
It is important to update the literature.
Author Response
We sincerely thank the reviewer for their valuable feedback and for acknowledging the manuscript's contribution to the generation of new knowledge. We have carefully considered each suggestion and have implemented the necessary revisions to enhance the quality and clarity of our manuscript:
- Introduction and Recent Literature: We agree that a robust introduction is crucial. We have thoroughly updated the Introduction section by incorporating relevant literature from the last five years to complement the problem statement and provide a more current context for our study. Regarding the "research questions resolved in the results," we clarify that this study adopted an exploratory design aimed at assessing psychometric equivalence and utility, rather than testing formal hypotheses. The objectives driving our analysis and the findings addressing them are clearly presented in the Results section.
- Methodology Specification: As suggested, the 'Materials and Methods' section now precisely specifies the design, approach, type, and timing of the study. It clearly states that this is a quantitative, cross-sectional, and correlational design, with synchronous data collection conducted at a single point in time.
- Sample Formation and Criteria: We have expanded the 'Participants' subsection within 'Materials and Methods' to provide a detailed explanation of how the sample was formed. This includes specifying the non-probabilistic convenience sampling method, justifying its choice within the university setting, and explicitly listing both the inclusion and exclusion criteria for participation.
- Strengthening the Discussion: The Discussion section has been strengthened by aligning its content with the updated Introduction, which now includes recent studies. This ensures a cohesive narrative and reinforces the contemporary relevance of our findings in the broader scientific landscape.
- Improved Conclusions and Novelty: We fully agree with the importance of highlighting the unique contributions of our study. We have revised the 'Conclusions' section to explicitly emphasize the novelties contributed by our research, particularly in the validation of slider-based assessment items within a mobile application for a comprehensive range of suicide-related dimensions, and how these findings compare to existing literature.
- Literature Update: Throughout the manuscript, we have undertaken a comprehensive update of the literature, ensuring that references are current and reflect the latest advancements in the field.
We believe that these revisions significantly improve the manuscript and address all the reviewer's valuable points.
Round 2
Reviewer 2 Report
Comments and Suggestions for Authors
The authors have addressed most of the comments in the previous review. The authors revised some sections. Some required appropriate references, such as in Lines 198-204, 282-286, 365-368, etc.
Author Response
The authors have addressed most of the comments in the previous review. The authors revised some sections. Some required appropriate references, such as in Lines 198-204, 282-286, 365-368, etc.
Dear Reviewer,
We sincerely appreciate your valuable comments and suggestions for the improvement of our manuscript. We have carefully reviewed each of them and are pleased to inform you that we have addressed most of the observations raised in the previous review.
We have conducted a thorough revision of several sections, incorporating the suggested changes. In particular, we have added the appropriate references in lines 198-204, 282-286, and 365-368, among others, to strengthen the empirical and theoretical basis of our statements.
We hope these changes meet your expectations and that the manuscript is now more robust.
Reviewer 3 Report
Comments and Suggestions for Authors
Dear Editor,
Thank you for the opportunity to review the revised version of the manuscript. The authors have responded constructively to the comments previously raised and have implemented changes that, in my view, have improved the overall quality of the work.
I believe the revised manuscript is now suitable for publication in Behavioural Sciences.
Kind regards,
Jan Francesco Arena, M.D.
Sapienza, University of Rome
Author Response
Dear Editor,
Thank you for the opportunity to review the revised version of the manuscript. The authors have responded constructively to the comments previously raised and have implemented changes that, in my view, have improved the overall quality of the work.
I believe the revised manuscript is now suitable for publication in Behavioural Sciences.
Kind regards,
Jan Francesco Arena, M.D.
Sapienza, University of Rome
Response:
We are very pleased to hear that the reviewer believes our revisions have improved the overall quality of the manuscript and that it is now considered suitable for publication in Behavioural Sciences. We greatly appreciate their constructive input and positive assessment.
Reviewer 4 Report
Comments and Suggestions for Authors
The authors have updated the introduction section, taking care to specify the design, approach, type, and timeline of the study. In this version of the manuscript, they indicate that it is a quantitative, cross-sectional, correlational design, with synchronous data collection at a single point in time. They explain in detail how they selected the sample. They improved the discussion and included in the conclusions the scientific contributions of the study and the new findings compared to similar studies.
Therefore, the manuscript should be published in its current version.
Author Response
The authors have updated the introduction section, taking care to specify the design, approach, type, and timeline of the study. In this version of the manuscript, they indicate that it is a quantitative, cross-sectional, correlational design, with synchronous data collection at a single point in time. They explain in detail how they selected the sample. They improved the discussion and included in the conclusions the scientific contributions of the study and the new findings compared to similar studies.
Therefore, the manuscript should be published in its current version.
Response:
We are delighted to learn that the reviewer recognizes the efforts made to update the introduction, specifying the study's quantitative, cross-sectional, correlational design, and synchronous data collection. We are also pleased that the detailed explanation of our sample selection, the improved discussion, and the inclusion of the study's scientific contributions and new findings in the conclusions are well-received.
We are very grateful for this assessment and the recommendation for publication in Behavioural Sciences.